# Assessing Mild Traumatic Brain Injury-Associated Blood–Brain Barrier (BBB) Damage and Restoration Using Late-Phase Perfusion Analysis by 3D ASL MRI: Implications for Predicting Progressive Brain Injury in a Focused Review

**DOI:** 10.3390/ijms252111522

**Published:** 2024-10-26

**Authors:** Charles R. Joseph

**Affiliations:** Department of Neurology and Internal Medicine, College of Osteopathic Medicine, Liberty University, Lynchburg, VA 24502, USA; crjoseph@liberty.edu

**Keywords:** mild traumatic brain injury, chronic traumatic encephalopathy, blood–brain barrier dysfunction, shear white matter injury, arterial spin labeling MRI, diffusion tensor imaging MRI, 18-fluorodeoxyglucose positron emission tomography, dynamic contrast-enhanced (DCE) MR perfusion

## Abstract

Mild traumatic brain injury (mTBI) is a common occurrence around the world, associated with a variety of blunt force and torsion injuries affecting all age groups. Most never reach medical attention, and the identification of acute injury and later clearance to return to usual activities is relegated to clinical evaluation—particularly in sports injuries. Advanced structural imaging is rarely performed due to the usual absence of associated acute anatomic/hemorrhagic changes. This review targets physiologic imaging techniques available to identify subtle blood–brain barrier dysfunction and white matter tract shear injury and their association with chronic traumatic encephalopathy. These techniques provide needed objective measures to assure recovery from injury in those patients with persistent cognitive/emotional symptoms and in the face of repetitive mTBI.

## 1. Introduction

Mild traumatic brain injury (mTBI) is a pervasive problem worldwide and typically managed without hospitalization thus unreported. The estimated prevalence rate of 369 per 100,000 individuals underestimates the scope of the problem as a result [1,2,3]. Most head injuries are mild (90%) defined by clinical functionality scales such as the Glasgow coma Scale where a score of 13–15/15 is defined as mild [4]. The scale includes basic tests of the level of consciousness as defined by verbal, eye opening, and motor responses to commands or noxious stimuli. It is an invaluable tool for assessing a TBI and for follow-up of moderate or severe TBI. More subtle brain injury, however, is not captured by this scale, as it is meant as a simple stratification tool for the rapid assessment and prognosis of more serious head trauma [4].

The assessment of mTBI is currently the sole purview of clinical assessment to categorize acute cognitive or other subtle neurologic deficit, along with behavioral/emotional effects [5]. Likewise, return to normal learning or sport activities are based on these measures [6]. Since anatomic imaging studies in mTBI are almost invariably normal, they are often not employed [7,8,9,10]. However, developing a non-invasive physiologic measure of acute injury and subsequent recovery would provide a critical objective assessment of a safe return to sport/usual activities.

### 1.1. Background

#### 1.1.1. Minor Closed Head Injury

Mild traumatic brain injury (mTBI) is defined as a traumatic brain injury resulting in brief if any loss of consciousness with associated transient cognitive (thinking, memory, general confusion), neurologic (vertigo, delayed reaction times), and/or behavioral (depression, aggression) dysfunctions, and often headache [11]. The most frequent causes include contact sport, blast, and acceleration–deceleration injuries. Sports injuries are acutely assessed on the field acutely or within 72 h using validated clinical testing instruments such as the Sport Concussion Assessment Tool (SKAT 6), Glasgow coma scale, and Vestibular Ocular Motor Screening (VOMS) [5,12,13], along with inclusion of the testing measures of cognitive function (orientation, immediate and delayed memory recall, concentration, emotionality, depression, anxiety, speech, and task specific processing, as well as coordination, balance, eye movements, and headache) [14]. A more comprehensive version for office use, SCOAT6, has also been validated for use within 72 h of injury. In addition to testing the SKAT 6 items, it catalogues prior head injuries, cognitive and emotional history, current medications, and a detailed neurologic examination. Further, it includes recommended strategies for a return to learning and return to sports activities [15,16]. Currently, corroborating physiologic testing is not commonly employed.

To that end, the physiologic consequences of an mTBI will be reviewed, and methods to identify those changes using imaging modalities will be discussed. The potential combined use of both clinical and physiologic testing may improve the accuracy of diagnosis and readiness to return to usual activities and predict persistent physiologic dysfunction. The latter when associated with prior mTBIs may potentially lead to chronic progressive disease—chronic traumatic encephalopathy (CTE)—years later, as seen in professional contact sports athletes [17].

#### 1.1.2. Pathophysiology

Given the brain is a soft delicate organ contained within a closed box, sudden applied blunt trauma and inertial forces of acceleration, deceleration, or rotational trauma dictate the severity and location of cortical, white matter tract, and vasculature injury. Unlike moderate or severe TBI, mTBI injury is associated with minimal or no disruption of these structures in most cases, but not all [18,19]. That said, the physiologic consequences of disrupted electrolyte pumps and blood flow acutely post head injury have been well delineated with an acute impairment of electrolyte (sodium potassium, calcium, and magnesium) distribution, notably mitochondrial calcium sequestration with altered metabolism in neurons and astroglia and reduced local blood flow [20,21,22]. This results in an acute glycolysis hypermetabolism to restore normal electrolyte balance [22]. Disruption of the blood–brain barrier (BBB) even in mild traumatic brain injury (mTBI) is reflective of this acute phase of injury [22,23,24,25,26]. The duration of this varies, dependent upon location and the severity of injury, leading to hypometabolism in the affected regions which can persist for weeks or longer [23,25,26]. The chronic phase of injury is largely thought to be related to persistent physiologic alterations from persistent activated microglia and their effect on the neurovascular unit [24,25,27,28,29]. The focus of this article principally addresses the BBB dysfunction and identifying imaging techniques. Additionally, techniques demonstrating shear force white matter tract injury will be discussed.

#### 1.1.3. Shear Injury in TBI

The most troublesome and unfortunately permanent sequela of TBI is the result of shear injury to white matter tracts [18,29,30]. The extent and severity of damage escalates proportional to the force and location of impact with compiled additional effects from the edema mass effect and inflammatory upregulation. In mTBI, the general assumption is an absence of substantive anatomic injury [17,18]. This, however, belies subtle changes in personality and emotionality, particularly in adolescence, from damage to developing limbic connections [31,32,33]. 

The underpinnings of shear injury are quite understandable with sudden mechanical deformation of the brain tearing apart connections [26,27,28,29,33,34]. The longer-term damage though is caused by the subsequent mass effect from edema, hemorrhage, or later from inflammation [22,26,27,28,29,34,35,36,37,38,39,40]. To understand the latter influences, animal models have been employed using a variety of species, most commonly rodents and, rarely, large animals [37,38,39]. Translatability from rodent to human physiology is problematic given the vast differences in brain structure and physiologic responses to injury [39,40]. 

The current medical state of the art possesses no means of restoring shear tract injury. The prevention of delayed injury by reducing the mass effect and swelling preserve life and may reduce secondary shear injury [28,30,31,32,33]. But, knowledge of how best to limit the secondary tract damage will require additional mechanistic study in higher animals (swine or primates) with a closer anatomic affiliation to the human brain [39,40].

Too often, the die is cast for potential clinical recovery based on the initial extent of shear force injury [37,38,39,40]. That said, identifying the severity of tract injury has potential importance in prognostic discussions with the patient and family.

#### 1.1.4. Blood–Brain Barrier Dysfunction in Acute and Chronic TBI

In mTBI, the microcirculation (capillary system) BBB is disrupted acutely, causing reduced mean capillary transit time (cMTT) and reduced glymphatic flow (GF) in the region(s) of impact both coup and contrecoup with limited or no white matter tract injury [36,37,38]. The effect of this disruption is leakage and the trapping of normally restricted substances into the brain and thus reduced clearance out of the brain. This results in a measurable reduction in the clearance rate of intraparenchymal fluid which can be identified by imaging techniques (see below). In animal studies, BBB disruption and leakage of Immunoglobulin G IGG locally persisted for up to a month following moderate TBI with associated delayed microhemorrhages at those sites [19,34,37,38,39]. 

A brief summary of the complexity of the BBB structure and function is presented here. The details of the cellular interactions, signaling mechanisms, and what is known of the basic biology are beyond the scope of this review but are referenced [24,41,42,43,44,45,46,47,48,49,50,51]. 

The blood–brain barrier is dependent on a complex interplay of cells within the neurovascular unit (NVU), including the endothelial cells, pericytes, and astrocyte end feet [34,43,44,45,50]. Signaling within this triad in homeostasis allows for the transport of needed electrolyte, glucose, and specific lipids and proteins while excluding common blood constituents that are toxic to the complex interstitial environment [41,42,43,44,45,46,47,48,49,50]. The interaction and communication among elements of the NVU, extracellular matrix (ECM), and microglia dictate the presence, expression, and integrity of the various tight junction (TJ) proteins [52,53]. The presence of specific inflammatory cytokines may result in either a proinflammatory milieux or, counterintuitively, a restorative one [25,54,55,56,57,58]. For example, endothelial cell-derived IL1 β acutely downregulates the TJ protein ZO-1/occludens, thus increasing the BBB permeability, but by promoting the expression of Pentraxin 3 it later enhances its restoration [57]. The relative balance of local microglial proinflammatory versus homeostatic regulatory influences also determines the BBB integrity [59,60,61,62,63]. That balance is dictated by neuronal and astrocyte signaling which is influenced by the presence of cell injury and local inflammation [43,60]. 

Fundamental questions remain regarding defining the restorative pathways post injury and the long-term consequences of persistent leakage [59,60,61,62,63]. In TBI, BBB leakage is caused by proinflammatory microglial–endothelial signaling and to shear forces disrupting the endothelial glycocalyx, the latter allowing the leakage of IGG and the upregulation of inflammatory cytokines causing the local conversion of microglia to a proinflammatory state [51,64,65,66]. The activated microglia express the complement (C3a) fragment which in turn upregulates C3aR in capillary endothelial cells, altering their phenotype to a proinflammatory/immune cell attractant state with associated disruption of intercellular tight junctions, thus triggering additional BBB leakage [67]. In the chronic phase of injury, the presence of activated microglia induces BBB dysfunction, local chronic inflammatory changes, and neuronal dysfunction [23,35,65]. The persistence of this especially with multiple mTBI results in the perivascular accumulation of pTau within astroglia and neurons in the sulcal depths, the hallmark of CTE [65,66,67,68].

Restoration of normal BBB integrity following mTBI generally occurs quickly in youth; however, in some, the persistence of activated microglia can persist for months or longer resulting in a reduction in the threshold for additional mTBI-related pathology with clinical accompaniment [68,69,70]. With advancing age, recovery is less complete and the progression of both inflammation with BBB leakage and cognitive decline following TBI is more likely [71,72,73,74]. Further, the lower threshold and persistence of mTBI in young women is well recognized and postulated to be influenced by sex hormones and reduced muscle mass compared to their male counterparts [75,76]. The mechanisms of the BBB repair process in humans remain incompletely delineated, in particular the specifics of the pathways active in youth but lost in the circumstances of repetitive TBI or in the normal aging process [66,70,71,72,73]. 

The cumulative effects of multiple mTBI events cause ongoing inflammatory upregulation resulting in both permanent microvascular changes and a persistent upregulation of inflammation within the neuropil [61,65,74]. This induces altered synthesis, degradation, accumulation, and transsynaptic spread of toxic misfolded proteins, predominantly hyperphosphorylated Tau (pTau), but also Lewy body proteins, TDP-43, and to a delayed and minor degree β amyloid [61,74]. The consequence is the well-described high incidence of chronic traumatic encephalopathy (CTE) in professional contact sports athletes [77,78,79,80,81]. The clinical correlates include aggression, depression, suicide ideation, impulsivity, cognitive decline, and parkinsonian features [69,77,78,79,80,81]. To reduce the likelihood of progressive dementia from cumulative brain injury, the addition of objective physiologic testing in mTBI should prove invaluable in conjunction with clinical assessment to identify the early chronic inflammatory conversion.

Given the importance of maintaining BBB integrity, there is likely more than one pathway regulating restoration post injury [66,71]. That said, the duration and speed of clinical recovery are correlated with the patient’s age, location and magnitude of force, and prior head injuries [30,31,32,72,73]. The inverse correlation of the severity of the head injury and the restoration of BBB integrity is understandably related to anatomic and vascular disruption [73,77]. The caveat to this general rule in adolescence has been shown via clinical evidence of emotional dysfunction even in mTBI which can be persistent, suggesting limbic system tract injury [68,73,76]. 

Of particular interest is why these mechanisms rapidly repair the BBB leakages in young patients with a single mTBI but flag in the aging brain even without injury. The answer may lie in the long-term presence of proinflammatory microglia and ECM constituent protein alterations which increase local BBB permeability [70,71,72,73]. Alternatively, the upregulation of inflammatory cytokines related to other unrelated health issues may contribute [79,80,81,82]. Conversely, is it a failure of the normal restorative signaling pathways lost in the aging process that are the root cause? Or perhaps a combination all three mechanisms? This question has profound implications for treatment of the underpinnings of other diseases such as neurogenerative diseases where the loss of BBB integrity is the nexus of their development [63,79,80,81,82,83,84]. Further, of the two pathologic features of traumatic brain injury, the restoration of BBB integrity as opposed to shear injury appears to be the most amenable to intervention.

To assess the presence of BBB leakage, physiologic imaging methods are required. Likewise, the assessment of future treatment modalities to monitor outcomes of clinical treatment trials addressing BBB repair requires the same [71,78,85,86]. In the next sections, the current most usable and clinically available methods of anatomic and physiologic imaging will be discussed along with their roles in identifying specific pathologic features of head injury.

### 1.2. Structural Imaging in mTBI

Although anatomic imaging in singular mTBI is infrequently conducted and if performed is almost invariably normal, it clearly has a place in the evaluation of patients with multiple mTBIs and of course more severe injury. In moderate or severe acute brain injury, CT head is the study of choice for the evaluation of bleeding or mass effect [18,86,87]. In evaluating subacute or chronic injury, using the benefit of MRI capabilities for defining atrophy with T1 sequences, white matter lesions with fluid attenuated inversion recovery (FLAIR), and prior micro or macro hemorrhages with susceptibility weighted images (SWIs) are the gold standards [85,86,87]. 

### 1.3. Physiologic Imaging

The assessment of TBI potentially takes three forms: anatomic, clinical cognitive/neurologic examination, and physiologic testing. In mTBI, structural evaluation with CT or MRI is nearly always normal, whereas the clinical cognitive assessments are far more revealing. However, subtle changes in processing, memory, personality, and emotional stability may miss subclinical physiologic injuries. Current clinically available methodologies for assessing the physiologic change in BBB function include dynamic contrast-enhanced MRI (DCE-MRI) and arterial spin labeling (ASL) MRI [85]. The evaluation of WM tract shear injury can be assessed directly with MRI tractography–diffusion tensor imaging (DTI) [88]. Positron emission tomography (PET) assesses the reduction in glucose utilization reflective of a regional loss of normal metabolic function [22,85]. Indirect means include functional MRI imaging (not discussed further as it is not commonly used clinically). The pros and cons of each method will be discussed below.

#### 1.3.1. Diffusion Tensor Imaging

Diffusion tensor imaging is a non-invasive method of investigating white matter tract integrity, which leverages the imaging of water movement within the brain [88,89]. Water in the absence of obstruction will move randomly in all directions (isotropy). In the normal brain, however, water is forced to move parallel to white matter tract obstructions (anisotropy). With shearing force injury, these neat pathways are interrupted and hence the water present in those locations moves more isotopically. Diffusion imaging techniques can identify that disruption (Figure 1). Technical improvement to this methodology is under development. The current methods available lack sensitivity and are hampered by long scan times [89,90]. Nonetheless, as methods ameliorating persistent BBB leakage develop, knowledge of the extent of untreatable shear injury is of value in prognosticating ultimate neurologic recovery.

#### 1.3.2. Dynamic Contrast-Enhanced (DCE) MRI

Dynamic contrast-enhanced (DCE) MRI directly measures the leakage of contrast agent into the neuropil in excess of its natural infusibility. Given gadolinium contrast agents are mostly excluded from passage through the BBB normally, their passage increases with its disruption. The technique assesses the plasma volume and interstitial space and can calculate a volumetric transfer constant from vascular to interstitial space [92,93,94]. The excess accumulation of contrast in the interstitial space beyond the calculated diffusion amount is indicative of BBB leakage. This method allows the direct visualization of BBB leakage but is invasive (requires gadolinium contrast injection) and thus is not suitable for serial measurements over time, given the concern of retained gadolinium contrast within the CNS [92,93,94,95,96] (Figure 2). Specific software to sort out normal contrast diffusion through the BBB from that related to excess leakage is also required [93,94]. Depending on patient circumstances though, its use as a diagnostic one-off study when coupled with DTI may prove helpful in prognosticating recovery in those slow to do so. The combination would reveal the extent of BBB leakage and white matter tract injury.

#### 1.3.3. 18-Fluorodeoxyglucose Positron Emission Tomography (FDG-PET)

Utilizing radioactively labeled fluorodeoxyglucose (^18^FDG) and measuring the cellular uptake thereof via positron emission tomography (PET) indirectly measures metabolic activity [22,98,99]. With less metabolic activity, there is a reduced uptake of labeled glucose as occurs in acute mTBI [98,99]. In the acute phase of severe injury, there is an increased uptake of ^18^FDG corresponding to the increased glycolysis required to reestablish electrolyte balance [100,101]. In the chronic phase of injury, there is a reduced uptake of glucose corresponding to reduced local activity [101,102,103,104,105,106,107,108]. When overlayed on CT or MRI brain images, the localization of abnormalities is enhanced (Figure 3). Sensitivity, specificity, cost, and scan times limit its utility.

The application of tau PET scans will not be discussed here as they would not be employed in mTBI but in cases of repetitive TBI when the question of CTE arises. The reader is referred to excellent reviews of these techniques [95,106,107,108,110,111,112,113,114,115,116,117,118,119,120,121,122].

#### 1.3.4. Arterial Spin Labeling MRI 

Another method to indirectly determine BBB leakage utilizes the clearance rate of magnetically labelled protons using 3D ASL MRI technology. This technique leverages the physiologic effects of altered perfusion with delayed capillary mean transit time (cMTT) and reduced glymphatic outflow in the late phase of the perfusion cycle, correlating with the location of BBB leakage [123] (Figure 4). It is a non-invasive method using blood protons magnetically spin labeled in the neck and, after a specified post-labeling delay (PLD), residual signal averages are sampled in the desired cortical regions.

By using multiple sequential PLD’s in the late phase of the perfusion cycle, linear analysis can be employed to determine the rate of clearance (slope of the line) [123,125]. In regions of BBB leakage, there are localized changes in blood flow that correlate with diminished clearance rates as can be seen in preclinical dementia [126]. Since the signal measured is largely from trapped labeled water protons in the late phase of perfusion, it is much more sensitive in determining minor BBB leakages than gadolinium (large molecule)-based DCE [127]. 

Since this is a non-invasive technique, it is suitable for sequential studies without risk of exogenous contrast. Currently, when combined with FLAIR and susceptibility weighted images (SWIs), the scan time is about 18^1/2^ min [123]. ASL MRI requires a 3 Tesla magnet and is limited to a reduced S/N. The latter can be circumvented by measuring a large field of view and obtaining multiple PLDs. Using 3D data acquisition allows for multiple regions of the brain to be investigated in one study. For example, our usage in mTBI acquired clearance rates from six brain regions, homologous temporal, frontal, and parietal regions [123,125,126]. Newer ASL MRI sequences allow for acquiring multiple PLDs in one sequence, reducing scan times to under 5 min [127]. This technique has significant potential in following the progress of BBB repair or alternatively persistent leakage. In our pilot study of mTBI in college-age athletes, both reduced glymphatic clearance post-acute injury and the restoration of BBB integrity as measured by this technique correlated with high sensitivity with clinical recovery [125]. Future studies evaluating more serious head injury sequentially may shed light on those patients with persistent dysfunction which sets the stage for progressive dementia (CTE) [126,127]. In addition, this objective measure could help to determine when and if an injured athlete or soldier could safely return to their sport or active duty. The limitations of this technique include the indirect method of determination (the results are graphic in nature post analysis as opposed to specific images of leakage) and a low S/N ratio. The latter is compensated for by analyzing multiple data points and acquiring signal from a large field of view FOV in the region of interest (ROI) chosen for analysis. Given the test is non-invasive as well as time- and cost-efficient, this method is suitable for serial studies post TBI in the assessment of BBB dysfunction [63,125]. 

#### 1.3.5. Summary of Techniques

Table 1 below summarizes the imaging target, relative advantages, and disadvantages of the four physiologic imaging techniques presented in evaluating mTBI. They provide objective information regarding the state of brain and vascular function at the time they are employed post head injury. The techniques will continue to improve and become more widely available providing both prognostic information. They will also be essential as outcome measures in future treatment trials. 

Table 1 summarizes the DTI in identifying shear tract injury, which if present correlates with the persistence of neurologic deficit and recovery depending on its extent. DCE and ASL MRI identify the loss of BBB integrity: the former directly and the latter indirectly. ^18^FDG-PET measures metabolic activity and thus could be negatively influenced by either shear tract injury or BBB leakage. 

## 2. Conclusions

With the mounting evidence of severe late life repercussions (CTE) related to repetitive mTBI among professional athletes in contact sports, there is a need to evaluate the physiology leveraging the loss of BBB integrity post injury that, if persistent, correlates with progressive changes in CTE [39,60,79,85,128,129,130,131,132]. By adopting physiologic testing measures, a safe return to the sport could be more accurately predicted in mTBI. Since the potential of both subtle white matter tact shear injury and persistent BBB leakage occur simultaneously, the assessment of damage requires attention to both, which translates into white matter tract imaging as well as an assessment of BBB integrity [24]. Further, as we uncover the mechanisms of BBB repair going forward and can leverage that knowledge therapeutically, identifying and assessing the outcomes of future treatments will be possible. The benefits of identifying BBB dysfunction spill over from TBI to preclinical stages of neurodegenerative disease (NG), where the loss of its integrity is at the nexus of development and a prime target for early intervention.

## Figures and Tables

**Figure 1 ijms-25-11522-f001:**
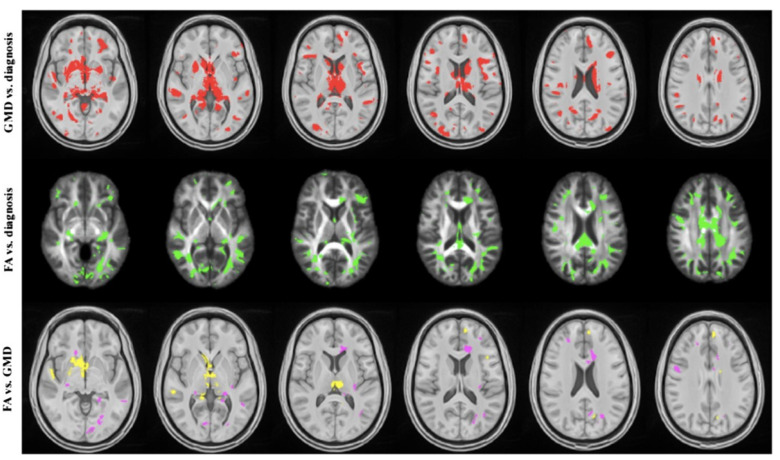
Sparse canonical correlation analyses (SCCA) of T1-weighted MP-RAGE and 30-direction diffusion tensor images (DTI) datasets are used to quantify the traumatically induced disruption of WM and cortical networks. The cohort includes 17 controls and 16 patients with TBI (age and gender matched). Each patient had a history of non-penetrating TBI of at least moderate severity. White matter integrity is assessed by DTI, and fractional anisotropy (FA) maps are generated. Separately, probabilistic segmentation of the T1-weighted imaging is performed to assess gray matter integrity. Variation in brain shape across subjects is normalized by diffeomorphically mapping these data into a population-specific template space. Image processing steps rely on the Camino and ANTs (Advanced Normalization Tools) neuroimage analysis open-source toolkits. SCCA demonstrates significant differences between the control and patient groups in both the FA (*p* < 0.002) and gray matter (*p* < 0.01) that are widespread but largely focus on thalamocortical networks related to the limbic system. Using SCCA identified regions, a strong correlation is identified between degree of injury in WM and GM within the patient group. Figure courtesy of James R. Stone. Reference [91]. Gandy et al., Reference [91]. © 2014 Gandy et al.; licensee BioMed Central Ltd. This is an open access article distributed under the terms of the Creative Commons Attribution License (http://creativecommons.org/licenses/by/2.0, accessed on 29 July 2024) which permits unrestricted use, distribution, and reproduction in any medium, provided the original work is properly credited. The Creative Commons Public Domain Dedication waiver (http://creativecommons.org/publicdomain/zero/1.0/ accessed on 29 July 2024) applies to the data made available in this article, unless otherwise stated.

**Figure 2 ijms-25-11522-f002:**
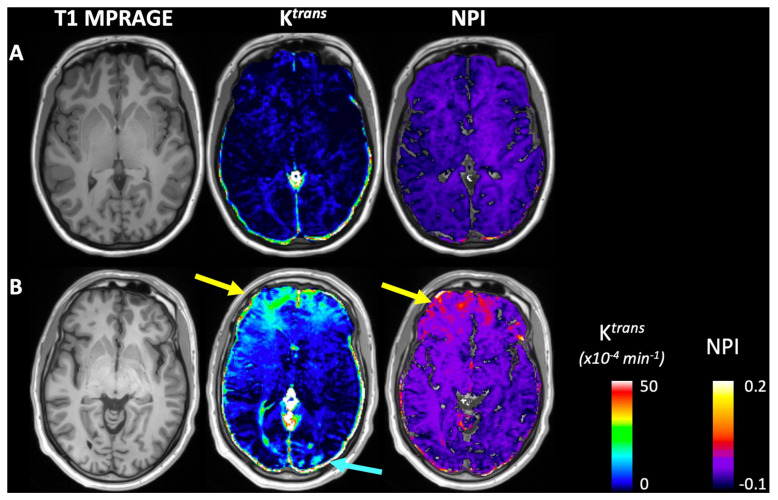
Individual subject examples from a representative control (**A**) and subacute TBI subject (**B**). The TBI non-lesional patient shows similar mild elevations in K*^trans^* and NPI in the anterior frontal white matter (yellow arrows) and additional small areas of K*^trans^* elevation in the occipital lobes (blue arrow). Ware et al., Ref [97]. The Authors. Published by Elsevier Inc. Amsterdam, The Netherlands. This is an open access article under the CCBY-NC-ND license (https://creativecommons.org/licenses/by-nc-nd/4.0/) accessed on 29 July 2024.

**Figure 3 ijms-25-11522-f003:**
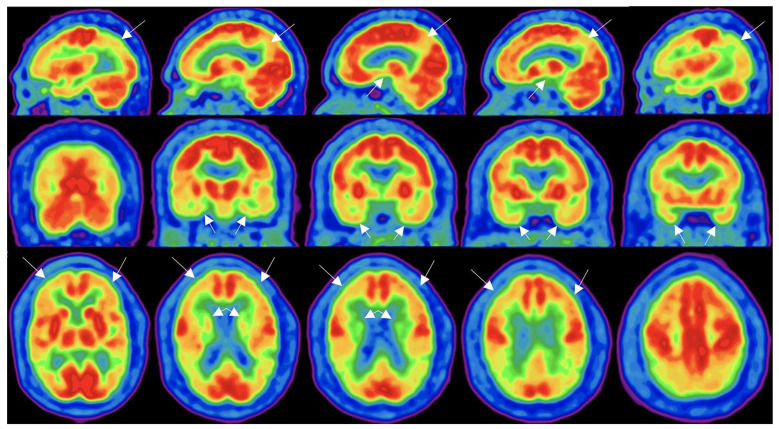
FDG-PET scan of the brain. (**Top**) Sagittal views (left to right) illustrate hypometabolism in the bilateral posterior parietotemporal regions, posterior cingulate, and precuneus. (**Middle**) Coronal views (posterior to anterior) show decreased uptake in the medial temporal lobes. (**Bottom**) Axial views (inferior to superior) demonstrate hypometabolism in the frontal lobe and anterior temporal poles. Morgan, et al. Ref [109]. This is an open access article under the CC BY-NC-ND license accessed on 29 July 2024.

**Figure 4 ijms-25-11522-f004:**
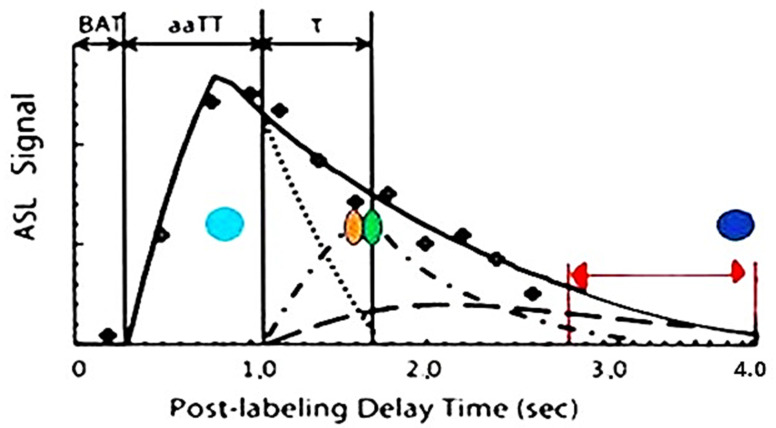
Composition of the ASL signal/time from its various components: arterial (dotted line), capillary (dot and dash line), and extra capillary (dashed line) spaces. The TI (acquisition) times utilized in this study are between the red arrow limits. Note that the signal composition is nearly completely extravascular (interstitial) water, which is normally cleared by the glymphatic system. The T1 times (63% signal decay) of major signal contributors are indicated by the colored dots. Magenta dot = 800–850 ms T1 of white matter. Orange dot = 1650 ms T1 of blood. Green dot = 1700 ms T1 of gray matter (all values are for 3T). Dark blue dot = 3800 ms T1 of water (CSF fluid); (all values are for 3T). BAT = bolus arrival time; aaTT = artery–artery arrival time; τ = peak capillary arrival time. Adapted with the publisher’s permission (Wiley) from reference [124].

**Table 1 ijms-25-11522-t001:** Methods of BBB and shear injury imaging modalities and limitations.

Imaging Procedure	Target	Sensitivity	Limitations
Diffusion Tensor Imaging (DTI) MRI	Diffusion of water-white matter tracts	Not clear, older techniques problematic; New techniques in development	Technique specific sensitivity, scan time
Dynamic Contrast-Enhanced (DCE) MRI	BBB leakage of contrast	Sensitive for BBB leakage	Requires gadolinium contrast, long scan times, not suitable for serial studies due to cerebral accumulation of GD^+^
^18^FDG-PET	Glucose utilization metabolic activity	Can identify the brief hypermetabolism acutely (animal studies) and chronic hypometabolism (human) chronically across severity spectrum	Cost, long scan times, availability of technology
3D ASL MRI	Perfusion/Diffusion of labeled protons	Highly sensitive in a small clinical trial in mTBI determining altered cMTT/BBB leakage with delayed BBB clearance/recovery	Requires a 3T or greater field strength, low signal, hand-drawn ROI, requires a larger series for confirmation of sensitivity

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
