# Peer review of "Assessing Mild Traumatic Brain Injury-Associated Blood–Brain Barrier (BBB) Damage and Restoration Using Late-Phase Perfusion Analysis by 3D ASL MRI: Implications for Predicting Progressive Brain Injury in a Focused Review"

_ijms, 2024, doi:10.3390/ijms252111522_

Round 1
Reviewer 1 Report
Comments and Suggestions for Authors
This is a focused review of imaging techniques available to investigate the integrity and functioning of the Blood blood-brain barrier (BBB) after mild traumatic brain injury (mTBI).
1. Clarity on the Role of BBB Disruption: The paper does not adequately address the debate on whether Blood-Brain Barrier (BBB) disruption is a causative of pathology or merely a marker for the severity of mild Traumatic Brain Injury (mTBI).
2. Line-by-Line Edits:
• Line 196: “FLAIR” should be capitalized for consistency.
• Line 206: There appears to be confusion between fMRI and PET. Clarification is needed to distinguish these imaging modalities.
3. Table Formatting and Captions:
• Table 1 lacks a proper caption, and the columns are not formatted properly. The table is not a summary of pros and cons but rather is a summary of methods for assessing the BBB after mTBI and their limitations. The caption should reflect this accurately.
4. Quality of Images: The images in the paper seem to be screenshots from other journals, leading to variations in quality and legibility. This detracts from the overall presentation. While this isn’t a deal-breaker, the authors should consider reformatting the data for consistency and clarity.
5. Title Recommendation: Since the paper is a literature review that lacks new data and is not a systematic review, it would be more accurate to include “focused review” or “selective review” in the title to reflect the scope and nature of the paper.
Author Response
For research article
|
Response to Reviewer 1 Comments
|
||
|
1. Summary |
|
|
|
Thank you very much for taking the time to review this manuscript. Please find the detailed responses below and the corresponding revisions/corrections highlighted/in track changes in the re-submitted files. [This is only a recommended summary. Please feel free to adjust it. We do suggest maintaining a neutral tone and thanking the reviewers for their contribution although the comments may be negative or off-target. If you disagree with the reviewer's comments please include any concerns you may have in the letter to the Academic Editor.]
|
||
|
2. Questions for General Evaluation |
Reviewer’s Evaluation |
Response and Revisions |
|
Does the introduction provide sufficient background and include all relevant references? |
Yes/Can be improved/Must be improved/Not applicable |
[Please give your response if necessary. Or you can also give your corresponding response in the point-by-point response letter. The same as below] |
|
Are all the cited references relevant to the research? |
Yes/Can be improved/Must be improved/Not applicable |
|
|
Is the research design appropriate? |
Yes/Can be improved/Must be improved/Not applicable |
|
|
Are the methods adequately described? |
Yes/Can be improved/Must be improved/Not applicable |
|
|
Are the results clearly presented? |
Yes/Can be improved/Must be improved/Not applicable |
|
|
Are the conclusions supported by the results? |
Yes/Can be improved/Must be improved/Not applicable |
|
For review article
|
Response to Reviewer 1 Comments
|
||
|
1. Summary |
|
|
|
Thank you very much for taking the time to review this manuscript. Please find the detailed responses below and the corresponding revisions/corrections highlighted/in track changes in the re-submitted files. I very much appreciate your careful review of this manuscript and have made all of your suggested edits.
|
||
|
2. Questions for General Evaluation |
Reviewer’s Evaluation |
Response and Revisions |
|
Is the work a significant contribution to the field? |
|
[Please give your response if necessary. Or you can also give your corresponding response in the point-by-point response letter. The same as below] |
|
Is the work well organized and comprehensively described? |
|
|
|
Is the work scientifically sound and not misleading? |
|
|
|
Are there appropriate and adequate references to related and previous work? |
|
|
|
Is the English used correct and readable? |
|
|
|
3. Point-by-point response to Comments and Suggestions for Authors |
|
|
|
Comments 1: Clarity on the Role of BBB Disruption: The paper does not adequately address the debate on whether Blood-Brain Barrier (BBB) disruption is a causative of pathology or merely a marker for the severity of mild Traumatic Brain Injury (mTBI). |
||
|
1. Response 1 I would argue that BBB disruption is both causative and a useful marker of dysfunction and recovery. Our pilot study (ref 116) is strong circumstantial evidence of the presence of disrupted BBB integrity with presence of cognitive disfunction in acute mTBI and subsequent BBB repair associated with clinical cognitive recovery. The introduction points to prolonged microglia related inflammatory effects on the interstitium even in a subset of mTBI set in motion by BBB dysfunction. That said the physiologic methods discussed do not in themselves prove cause effect relationship of BBB dysfunction and associated cognitive impairment. Nonetheless, they certainly can be extremely useful as objective measures to define recovery. This may reduce compounding injuries among athletes from premature return to their sport, or soldiers returning prematurely to active duty. In the future as methods of treatment of BBB dysfunction emerge in more severe TBI, they may serve as invaluable outcome measures. I have revised the manuscript to hopefully clarify your concern.
|
||
|
Comments 2: Line 196: “FLAIR” should be capitalized for consistency. • Line 206: There appears to be confusion between fMRI and PET. Clarification is needed to distinguish these imaging modalities.
|
||
|
Response 2: Agree. I have, accordingly, revised the document and clarified the fMRI/PET distinction. Thank you for catching that.
|
||
|
Comment 3 Table 1 lacks a proper caption, and the columns are not formatted properly. The table is not a summary of pros and cons but rather is a summary of methods for assessing the BBB after mTBI and their limitations. The caption should reflect this accurately.
|
||
|
Response 3. Thank you for catching that. I have revised the table and provided a caption. |
||
|
Comment 4 Quality of Images: The images in the paper seem to be screenshots from other journals, leading to variations in quality and legibility. This detracts from the overall presentation. While this isn’t a deal-breaker, the authors should consider reformatting the data for consistency and clarity.
|
||
|
Response 4. I have hopefully improved the image quality in my edits |
||
|
5. Title Recommendation: Since the paper is a literature review that lacks new data and is not a systematic review, it would be more accurate to include “focused review” or “selective review” in the title to reflect the scope and nature of the paper.
|
||
|
Response 5 Great idea…will incorporate focused review in the title |
||
|
|
||
|
5. Additional clarifications |
||
|
Have made the necessary changes to the figure 1 where the caption was included in the picture. |
||
Reviewer 2 Report
Comments and Suggestions for Authors
The manuscript is a review of MR imaging techniques used to identify blood brain barrier dysfunction and white matter tract shear injury, and their link to mild TBI. There are some major issues with this review.
1. Given that mTBI is so hard to diagnose, it is unclear how the mTBI causes the leakage of the BBB. What is the relevance of this to clinical practice? Are there treatments that could help?
2. It is unclear how the information provided by the MRIs could be directly linked with BBB leak. Could there be other reasons for what the MRIs are shown that may be related to the mTBI but not necessarily to a BBB leak? Without a proper presentation of the BBB structure and functions and the changes caused by mild/repetitive traumatic mechanical forces, it is difficult to say for sure that the BBB leak is a biomarker for mTBI. Are the MRIs correlated to other clinical methods that check for the BBB fitness? Repetitive mTBI could lead to TBI, so it may be possible that the MRIs could pick moderate TBI.
3. There are some relatively recent reviews of the BBB structure and functions that discuss imaging modalities. How do those relate to this work?
4. There are many acronyms, some not introduced (IGG, NVU). Once the full name and its acronym are given, then please use the acronym throughout the rest of the paper. A list of acronyms may also be useful.
5. There are minor issues with the English language. Please follow proper English grammar rules.
Comments on the Quality of English LanguageThere are issues with the English grammar that must be addressed.
Author Response
|
Response to Reviewer 2 Comments
|
||
|
1. Summary |
|
|
|
Thank you very much for taking the time to review this manuscript. Please find the detailed responses below and the corresponding revisions/corrections highlighted/in track changes in the re-submitted files. Your time and attention is greatly appreciated.
|
||
|
2. Questions for General Evaluation |
Reviewer’s Evaluation |
Response and Revisions |
|
Does the introduction provide sufficient background and include all relevant references? |
Yes/Can be improved/Must be improved/Not applicable |
[Please give your response if necessary. Or you can also give your corresponding response in the point-by-point response letter. The same as below] |
|
Are all the cited references relevant to the research? |
Yes/Can be improved/Must be improved/Not applicable |
|
|
Is the research design appropriate? |
Yes/Can be improved/Must be improved/Not applicable |
|
|
Are the methods adequately described? |
Yes/Can be improved/Must be improved/Not applicable |
|
|
Are the results clearly presented? |
Yes/Can be improved/Must be improved/Not applicable |
|
|
Are the conclusions supported by the results? |
Yes/Can be improved/Must be improved/Not applicable |
|
|
3. Point-by-point response to Comments and Suggestions for Authors |
||
|
Comments 1: Given that mTBI is so hard to diagnose, it is unclear how the mTBI causes the leakage of the BBB. What is the relevance of this to clinical practice? Are there treatments that could help? |
||
|
1. Response 1: mTBI as you know is a clinical diagnosis based on alterations of cognitive and neurologic function post minor head injury. At present there are no specific test markers to correlate with clinical assessment tools. As an exploratory method to assess altered late phase perfusion which assesses capillary filling and glymphatic flow, we used ASL MRI in college athletes post mTBI acutely. We found within a week of injury, there was reduced clearance of labeled blood constituent protons much as we found in patients with mild cognitive impairment and Alzheimer disease (see references 115, 116 ). Upon clinical recovery the clearance rate returned to normal in all the athletes. The relevance of employing this noninvasive methodology in clinical practice would provide an objective measure to assure the readiness of athletes to return to their sport, or soldiers to return to active duty, potentially reducing the risk of CTE from premature reentry and compounding injuries. Currently, the mechanism/pathways of BBB damage repair are incompletely understood. In youth repair with time is the general rule, provided there are not multiple compounding injuries. Since BBB dysfunction is inherent to multiple brain diseases including CTE, and neurodegenerative diseases, researching methods of enhancing or restoring BBB integrity is essential. That said, a reliable, sensitive, noninvasive and economical method such as ASL MRI to assess outcome of future treatments is essential.
|
||
|
Comments 2: It is unclear how the information provided by the MRIs could be directly linked with BBB leak. Could there be other reasons for what the MRIs are shown that may be related to the mTBI but not necessarily to a BBB leak? Without a proper presentation of the BBB structure and functions and the changes caused by mild/repetitive traumatic mechanical forces, it is difficult to say for sure that the BBB leak is a biomarker for mTBI. Are the MRIs correlated to other clinical methods that check for the BBB fitness? Repetitive mTBI could lead to TBI, so it may be possible that the MRIs could pick moderate TBI. |
||
|
Response 2. Mild TBI is diagnosed clinically with altered cognitive /balance function related to proximate head injury with brief or no loss of consciousness. There are no anatomic changes found typically on CT or routine MRI. That said the impact causes direct injury to the affected site with well described acute changes in BBB integrity and sets in motion the possibility of a delayed inflammatory injury to BBB as described in the paper. In our published article {ref. 116 of the manuscript}, we looked at a series of well-conditioned athletes without other proximate causes for cognitive function after a witnessed mTBI and studied the physiologic effect in the late phase of perfusion via PASL MRI. This technique clearly demonstrated reduction in clearance of labeled protons in localized injured regions of the brain acutely (within 7 days of injury), which cleared in association of clinical recovery. This technique provides a potential confirmatory diagnostic test of physiologic dysfunction and method of confirming clinical recovery. Since spontaneous recovery is nearly universal with a single mTBI injury no specific remedies outside of rest and avoidance of additional injury is required. Among athletes however, there is often “push” to return to sport activity before recovery is complete, with the risk of compounding injury setting in motion the inflammatory responses that in time may cause CTE (progressive neurodegenerative disease). Thus, if an objective measure of recovery is employed post mTBI, the risk of CTE may be reduced from compounding premature injuries. We have not studied moderate TBI but rather expect (untested) there would be persistence of the reduced glymphatic clearance rate for a longer or indefinite time. If that is borne out, then return to normal activity should be delayed until/if normal physiology is restored.
There are no other causes that can be reasonably attributed to either the cognitive or physiologic changes present acutely post mTBI or upon recovery in our mTBI study to explain the ASL MRI results documented in our published study. I refer you to the text and explanation of the physiologic basis of ASL MRI. The well described late phase perfusion ties to residual labeled fluid ties to residual signal in ASL MRI within the interstitium at long post labeling delay times of acquisition. It is composed of free labeled fluid from delayed capillary filling and leaked fluid within the interstitium that becomes trapped there due to impaired glymphatic clearance. The long T1 of residual fluid and lesser extent capillary flow account for the sum of residual signal present in the late phase of perfusion. The other tissues and blood have much shorter T1 values and hence minimal contribution to composite signal at the long post labeling data acquisition. In essence, with BBB leak, intravascular serum components leak into the interstitium and are trapped due to the simultaneous disruption in glymphatic clearance pathways. The end result is delayed rate of clearance of these fluids.
|
||
|
Comment 3. There are some relatively recent reviews of the BBB structure and functions that discuss imaging modalities. How do those relate to this work? |
||
|
Response 3. I have added a compendium review of MRI techniques to that point (Ref #118). My intent here was to provide clinically available methods of investigating head injury in community settings only. Investigative or difficult time consuming or overly expensive methods were purposely excluded, as they will unlikely be utilized.
|
||
|
4. There are many acronyms, some not introduced (IGG, NVU). Once the full name and its acronym are given, then please use the acronym throughout the rest of the paper. A list of acronyms may also be useful. |
||
|
Response 4. I have defined them in the text but to your suggestion will provide a list…great idea!
|
||
|
5. There are minor issues with the English language. Please follow proper English grammar rules. |
||
|
Response 5. Thank you for bringing to my attention lapses in English. I have made multiple editorial changes that should improve readability. |
||
|
|
||
|
|
||
|
|
||

Round 2
Reviewer 2 Report
Comments and Suggestions for Authors
Unfortunately, the updated manuscript did not sufficiently address the issues mentioned in the first review. A review of the literature at this early stage when there is not enough evidence to support a link between mTBI and BBB dysfunction observed with medical images is misleading.
Author Response
New references 10-9-24
Reviewer 2 request for additional references.
Below are the newly included references with their numeric location in the paper. Thanks
- Abrahamson, Eric E., and Milos D. Ikonomovic. "Brain injury-induced dysfunction of the blood brain barrier as a risk for dementia." Experimental Neurology328 (2020): 113257.
- 25. O'Keeffe, Eoin, Eoin Kelly, Yuzhe Liu, Chiara Giordano, Eugene Wallace, Mark Hynes, Stephen Tiernan et al. "Dynamic blood–brain barrier regulation in mild traumatic brain injury." Journal of neurotrauma37, no. 2 (2020): 347-356.
26.Elder, Gregory A., Miguel A. Gama Sosa, Rita De Gasperi, James Radford Stone, Dara L. Dickstein, Fatemeh Haghighi, Patrick R. Hof, and Stephen T. Ahlers. "Vascular and inflammatory factors in the pathophysiology of blast-induced brain injury." Frontiers in neurology 6 (2015): 48.
- Ojo, Joseph O., Benoit Mouzon, Moustafa Algamal, Paige Leary, Cillian Lynch, Laila Abdullah, James Evans et al. "Chronic repetitive mild traumatic brain injury results in reduced cerebral blood flow, axonal injury, gliosis, and increased T-tau and tau oligomers." J Neuropathol Exp Neurol75, no. 7 (2016): 636-655.
- Abdul-Muneer, P. M., Namas Chandra, and James Haorah. "Interactions of oxidative stress and neurovascular inflammation in the pathogenesis of traumatic brain injury." Molecular neurobiology51 (2015): 966-979.
- Kempuraj, Duraisamy, Mohammad Ejaz Ahmed, Govindhasamy Pushpavathi Selvakumar, Ramasamy Thangavel, Arshdeep S. Dhaliwal, Iuliia Dubova, Shireen Mentor et al. "Brain injury–mediated neuroinflammatory response and Alzheimer’s disease." The Neuroscientist26, no. 2 (2020): 134-155.
- 8 Li, Liangping, Jiawen Liang, and Hongjun Fu. "An update on the association between traumatic brain injury and Alzheimer’s disease: Focus on Tau pathology and synaptic dysfunction." Neuroscience & Biobehavioral Reviews120 (2021): 372-386.
- Griffin, Allison D., L. Christine Turtzo, Gunjan Y. Parikh, Alexander Tolpygo, Zachary Lodato, Anita D. Moses, Govind Nair et al. "Traumatic microbleeds suggest vascular injury and predict disability in traumatic brain injury." Brain142, no. 11 (2019): 3550-3564.
- Haarbauer-Krupa, Juliet, Mary Jo Pugh, Eric M. Prager, Nicole Harmon, Jessica Wolfe, and Kristine Yaffe. "Epidemiology of chronic effects of traumatic brain injury." Journal of neurotrauma38, no. 23 (2021): 3235-3247.
- Snowden, Taylor M., Anthony K. Hinde, Hannah MO Reid, and Brian R. Christie. "Does mild traumatic brain injury increase the risk for dementia? A systematic review and meta-analysis." Journal of Alzheimer's disease 78, no. 2 (2020): 757-775.